# Loss of Y chromosome in Alzheimer's patients co-occurs with somatic mutations beyond CHIP drivers

Edyta Rychlicka-Buniowska[1], Daniil Sarkisyan[2], Monika Horbacz[1], Bożena Bruhn-Olszewska[2], Kinga Drężek-Chyła[1], Mikołaj Koszyński[1], Hanna Davies[2], Ulana Juhas[1,3], Magdalena Wójcik-Zalewska[1], Alicja Klich-Rączka[4], Lena Kilander[5], Martin Ingelsson[5,6,7], Karolina Bukowska-Strakova[8], Kazimierz Węglarczyk[8], Maciej Siedlar[8], Jarosław Baran[8], Janusz Ryś[9], Arkadiusz Piotrowski[1], Natalia Filipowicz[1,*], Jan P Dumanski[1,2,,*]

**Loss of Y chromosome (LOY) and clonal hematopoiesis of indeterminate potential (CHIP) are common age-related events with implications for aging and Alzheimer disease (AD). LOY is linked to increased AD risk, whereas CHIP may be protective, and their co-occurrence remains unclear. We conducted whole-exome sequencing of CD4+ T cells, NK cells, and myeloid cells from AD patients and controls exhibiting LOY or retention of Y chromosome. We identified 39 variants in known myeloid and lymphoid driver genes, with up to 35% co-occurring with LOY in the same clone. In addition, we detected 192 unknown drivers of clonal hematopoiesis, enriched in AD-LOY individuals (odds ratio 4.8, _P_ = 0.041). In myeloid cells, total driver burden correlated with LOY (_ρ_ = 0.52, _P_ = 0.00041). These results indicate that LOY is a primary driver of clonal hematopoiesis in AD, seeding myeloid clones that accumulate unknown driver variants, whereas most canonical CHIP mutations arise independently. Our study reveals distinct, partially overlapping clonal architectures for LOY and CHIP and highlights LOY-driven myeloid expansion as a contributor to AD pathogenesis.**

## Introduction

Aging is accompanied by an accumulation of post-zygotic (somatic) mutations, particularly in hematopoietic stem cells (HSCs), which are estimated to acquire up to 0.858 exonic mutations per year, summing up to about 1.4 million protein-coding mutations in a 70-yr-old individual (Welch et al, 2012; Jaiswal & Ebert, 2019). When an HSC harboring such alterations gains a selective advantage, it gives rise to an expanded population of cells, in the process known as clonal hematopoiesis (CH). Clonal hematopoiesis of indeterminate potential (CHIP) specifically refers to CH with mutations accumulating in myeloid malignancy-associated genes (Steensma et al, 2015; Khoury et al, 2022; Vlasschaert et al, 2023). These CH/CHIP mutations are typically present in blood cells with a variant allele frequency (VAF) of at least 2% in the driver genes. However, variants below this threshold can also have adverse outcomes (McKerrell et al, 2015). Both CH and CHIP refer to individuals who do not meet diagnostic criteria for hematologic malignancies. CHIP is not only associated with higher risk of acute myeloid leukemia (Genovese et al, 2014; Jaiswal et al, 2014), but also with cardiovascular disease, all-cause mortality (Jaiswal et al, 2017; Nachun et al, 2021), and chronic kidney disease (Dawoud et al, 2022).

Loss of Y chromosome (LOY) is the most common post-zygotic aberration, occurring in normal hematopoietic cells (Forsberg et al, 2017; Dumanski et al, 2021) and exhibiting temporal dynamics (Danielsson et al, 2020; Bruhn-Olszewska et al, 2022). Hematopoietic LOY is detected in ~20% of men aged 40–70 (Thompson et al, 2019) and increases with age—from 5% in men younger than 50 yr to 40% in those older than 65 yr, reaching ~60% by 93 yr (Forsberg et al, 2019). LOY has also been found in other tissues, albeit at substantially lower frequencies (Haitjema et al, 2017; Abdel-Hafiz et al, 2023; Stańkowska et al, 2024). Initial reports correlated LOY to all-cause mortality, elevated risk of non-hematologic cancers and Alzheimer disease (AD) (Forsberg et al, 2014; Dumanski et al, 2016). Subsequent studies have linked it to a broader spectrum of disorders (summarized in Table 1 of Bruhn-Olszewska et al [2025]).

Since CHIP and LOY are common and age-related clonal events, investigating their co-occurrence and potential interaction emerged as a natural and important direction to understand age-associated clonal dynamics. LOY in bone marrow CD34+ cells was

---

[1]3P-Medicine Laboratory, Medical University of Gdańsk, Gdańsk, Poland   [2]Department of Immunology, Genetics and Pathology and Science for Life Laboratory, Uppsala University, Uppsala, Sweden   [3]Department of Bioenergetics and Physiology of Exercise, Medical University of Gdańsk, Gdańsk, Poland   [4]Department of Internal Medicine and Gerontology, Jagiellonian University Medical College, Kraków, Poland   [5]Department of Public Health and Caring Sciences/Geriatrics, Uppsala University, Uppsala, Sweden   [6]Krembil Brain Institute, University Health Network, Toronto, Canada   [7]Tanz Centre for Research in Neurodegenerative Diseases, Departments of Medicine and Laboratory Medicine and Pathobiology, University of Toronto, Toronto, Canada   [8]Department of Clinical Immunology, Institute of Pediatrics, Jagiellonian University Medical College, Kraków, Poland   [9]Department of Tumor Pathology, Maria Skłodowska-Curie National Research Institute of Oncology, Kraków, Poland

Correspondence: edyta.rychlicka-buniowska@gumed.edu.pl; jan.dumanski@igp.uu.se
*Natalia Filipowicz and Jan P Dumanski contributed equally to this work

**Table 1.  Basic cohort characteristics.**

| Variable | AD | | | CTRL | | |
|---|---|---|---|---|---|---|
| | **LOY** | **ROY** | *P*-value | **LOY** | **ROY** | *P*-value |
| (A) | | | | | | |
| N | 32 | 34 | — | 23 | 29 | — |
| Age | 83.5 ± 9.5 | 81.0 ± 7.0 | 0.22 | 72.0 ± 8.0 | 72.0 ± 5.0 | 0.84 |
| Current smoker | 0 | 1 | — | 5 | 5 | — |
| % LOY-CD4 | 1.1 ± 3.4[a] | 0.6 ± 5.0 | 0.15 | 2.4 ± 4.2[a] | 1.1 ± 2.1 | 0.35 |
| % LOY-NK | 16.5 ± 13.0 | 0.3 ± 3.8 | $P < 0.001$ | 16.5 ± 32.8 | 0.9 ± 3.9 | $P < 0.001$ |
| % LOY-MYEL | 44.4 ± 39.7 | 2.5 ± 5.1 | $P < 0.001$ | 36.8 ± 50.6 | 2.5 ± 5.0 | $P < 0.001$ |

| Cell types | AD | | CTRL | |
|---|---|---|---|---|
| | **LOY** | **ROY** | **LOY** | **ROY** |
| (B) | | | | |
| CD4/NK/MYEL | 32 | 32 | 18 | 25 |
| CD4/NK | — | 2 | 1 | 3 |
| CD4/MYEL | — | — | 4 | 1 |

**(A)** Number of subjects (N), age, smoking status and percentage of cells with LOY (%LOY) in the analyzed cohort; **(B)** Whole exome sequencing (WES) data from specific cell types. Age and %LOY in CD4[+] T, NK, and myeloid cells were given as median ± interquartile range; the Mann-Whitney *U* test was used to test the differences between the distribution of these variables in LOY and ROY subjects; CD4/NK/MYEL—number of subjects with WES data from CD4[+] T, NK and myeloid cells; CD4/NK—number of subjects with WES data from CD4[+] T and NK cells only; CD4/MYEL—number of subjects with WES data from CD4[+] T and myeloid cells only; CTRL, controls, AD, Alzheimer disease patients; LOY, loss of Y chromosome; %LOY, percent of cells lacking chromosome Y; ROY, retention of Y chromosome.
[a]CD4[+] T cells have low LOY levels even in subjects with high LOY levels in other cell types.

found in subjects with myelodysplastic syndrome (Ganster et al, 2015; Ouseph et al, 2021). In monocytes, LOY frequently co-occurs with pathogenic CHIP variants in individuals free of hematologic disorders (Ljungström et al, 2022). Furthermore, age-related CHIP in *TET2*, *TP53*, and *CBL* genes has been associated with LOY occurring at clonal fractions exceeding 30% (Dawoud et al, 2023), whereas large-scale population studies indicate that CHIP and LOY share germline genetic risk alleles (Kessler et al, 2022). Conversely, others describe reduced LOY in CHIP-positive individuals (Brown et al, 2023) or even no co-occurrence between LOY and CHIP (Kamphuis et al, 2023; Mas-Peiro et al, 2023).

Both CHIP and LOY are linked to immune dysregulation (Ferrucci & Fabbri, 2018; Dumanski et al, 2021; Cobo et al, 2022; Abdel-Hafiz et al, 2023; Mattisson et al, 2024; Wójcik et al, 2024), raising interest in their roles in neurodegenerative diseases. CHIP may confer neuroprotective effects by modulating systemic inflammation, potentially through altered microglial activity or other immune-related pathways slowing the progression of AD (Bouzid et al, 2023). Conversely, recent studies contradicted these findings, showing that CHIP has been associated with an increased risk of AD, especially in subjects with the *APOE ε3/ε3* genotype (Naito et al, 2025 *Preprint*; Choi et al, 2025 *Preprint*). Although these results have not yet been peer reviewed, they are consistent with the potential role of CHIP in other neurodegenerative disorders. For example, CHIP has been linked to increased risk of vascular neurodegenerative conditions and amyotrophic lateral sclerosis (Liu et al, 2024) with common driver mutations in *DNMT3A* and *TET2* concomitant with higher odds of developing Parkinson's disease (Woo et al, 2024) and multiple system atrophy (Lee et al, 2024). LOY has likewise

been implicated in AD pathogenesis, with evidence of its presence in leukocytes and brain cells (Dumanski et al, 2016; Caceres et al, 2020; García-González et al, 2023). The highest frequency of LOY-associated transcriptional changes in immune-related genes occurs in NK cells (Dumanski et al, 2016), whereas LOY-related alterations in DNA methylation have been reported in granulocytes and monocytes (Jąkalski et al, 2025). Notably, LOY has also been detected in microglia from AD patients' brains, suggesting its role in male-specific neurodegeneration (Vermeulen et al, 2022).

Given the accumulation of CHIP and LOY and their link to neurodegeneration, we set out to investigate their co-occurrence and potential pathogenic contribution to AD. We applied FACS to isolate subpopulations of lymphoid (CD4[+] and NK) and myeloid (granulocytes or monocytes) cells from AD patients and controls, stratifying individuals based on LOY or retention of Y chromosome (ROY). To analyze CH beyond the canonical CHIP driver genes, we performed high-coverage whole-exome sequencing (WES) on sorted fractions, interrogating 432 well-characterized myeloid and lymphoid driver genes alongside other protein-coding genes.

# Results

## The landscape of CH variants versus LOY status

Our aim was to analyze CH in peripheral leukocytes from lymphoid and myeloid lines collected from AD and CTRL individuals, stratified by LOY or ROY. Cell subpopulations with ≥15% of cells lacking

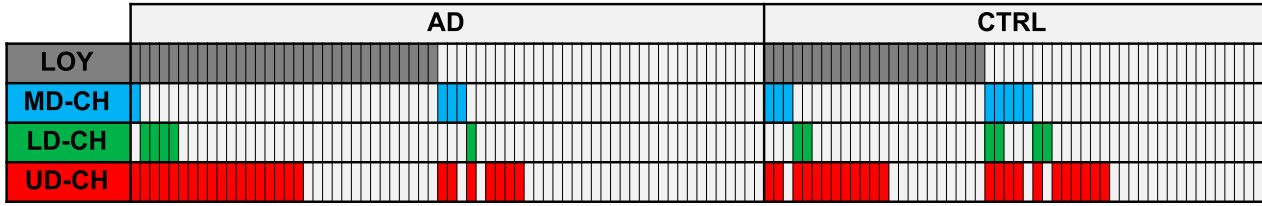

**Figure 1. Landscape of clonal hematopoiesis (CH) in 118 analyzed subjects.**
Each type of CH variants is stratified by the disease status (AD patients versus controls) and chromosome Y status (LOY versus ROY). Each column represents one subject. Each gray rectangle in the first row represents the LOY subject. Each following row represents one type of CH variants, and the presence of variants in subject is indicated by a specific color: blue—known myeloid driver gene (MD-CH), green—known lymphoid driver gene (LD-CH), and red—unknown driver gene (UD-CH). AD: n = 66 (32 LOY, 34 ROY); CTRL: n = 52 (23 LOY, 29 ROY); AD, Alzheimer disease; LOY, loss of Y chromosome; ROY, retention of Y chromosome; CTRL, control.

the Y chromosome were classified as LOY. Subjects with at least one cell type classified as LOY were assigned to the LOY group. Otherwise, subjects were assigned to the ROY group. CH was defined by post-zygotic single nucleotide variants (SNVs) and small indels, detected using WES with deep coverage (median on-target coverage 136, range 62–200; Table S1). Variants were classified as post-zygotic if detected in one or more, but not all, cell types from the same individual. Based on the affected genes, variants were categorized as myeloid drivers (MD-CH), lymphoid drivers (LD-CH), or unknown drivers (UD-CH). MD-CH and LD-CH variants affected genes associated with myeloid and lymphoid malignancies (Beauchamp et al, 2021; Niroula et al, 2021; Ljungström et al, 2022; Pich et al, 2022; Brown et al, 2023; Dawoud et al, 2023; Vlasschaert et al, 2023), respectively, whereas UD-CH comprised variants in all other protein-coding genes. The lists of the genes queried in this study are presented in the Tables S2 and S3, respectively.

The analyzed cohort included 66 AD patients (32 LOY, 34 ROY) and 52 CTRLs (23 LOY, 29 ROY) (Tables 1A and S4). Each subject contributed at least two cell types from lymphoid and myeloid lineages; 107 (90.7%) had all three fractions studied (Table 1B). In 193 samples, LOY level was estimated using mLRR-Y, for 133 samples we used ddPCR assay for *AMELY/AMELX* genes and 22 samples were analyzed using both methods, with ddPCR results preferred for LOY scoring (Table S4). Notably, the LOY levels were typically higher in myeloid and NK cells, whereas CD4$^+$ T cells showed lower levels even among LOY-classified subjects, which are consistent with previous findings (Dumanski et al, 2021).

The distribution of detected variants across the 118 analyzed subjects, stratified by disease status (AD versus CTRL) and Y chromosome status (LOY versus ROY), is shown in Fig 1 and Table S5. In total, 13 MD-CH variants (including one *SRSF2* variant occurring twice) were identified in 12 subjects (12/118; 10.2%)-four AD patients (4/66; 6.1%) and eight CTRLs (8/52; 15.4%). Most MD-CH variants were protein-truncating (9/13; 69%) and had high oncogenic potential based on COSMIC and/or ClinGen-CGC-VICC (Horak et al, 2022) annotations (12/13; 92%). The recurrently affected myeloid driver genes were *TET2*, *DNMT3A*, and *SRSF2* (Table 2A). No significant enrichment of MD-CH variants was observed in LOY subjects. LD-CH variants were detected in 11 subjects (11/118; 9.3%), including five AD patients (5/66; 7.6%) and six CTRLs (6/52; 11.5%), each carrying a unique variant, five of which (5/11; 45%) were protein-truncating. *KMT2D* was the only recurrently affected lymphoid gene and variants were exclusively observed in LOY

subjects (Table 2A). Furthermore, 192 UD-CHs were identified in 48 subjects (48/118; 40.7%), including 25 AD patients (25/66; 37.9%) and 23 CTRLs (23/52; 44.2%). Among these, 40 (40/192; 20.8%) variants were protein-truncating, detected in 26 individuals. There were no UD genes recurrently affected across subjects. UD-CHs were enriched in AD-LOY compared with AD-ROY patients (18/32, 56.2% versus 7/34, 20.6%, respectively; Fisher's exact test, Benjamini-Hochberg-adjusted *P* = 0.038). Overall, clonal hematopoiesis, defined by post-zygotic variants in any driver category (MD-CH, LD-CH, or UD-CH; collectively "Any-CH"), was detected in 52 individuals (52/118; 44%), including 26 AD (26/66; 39.4%) and 26 CTRL patients (26/52; 50%).

## The association between LOY and CH in AD patients and controls

To assess if different CH variants were enriched in LOY subjects, we stratified AD and CTRL donors based on the chromosome Y status. The proportions of subjects with MD-CH, LD-CH, UD-CH, and Any-CH in LOY and ROY groups were compared using logistic regression adjusted for age and age$^2$. There was no significant association between LOY and MD-CH or LD-CH (Fig 2A and B). However, UD-CHs (OR = 4.75, adjusted *P* = 0.041) and Any-CH variants (OR = 4.29, adjusted *P* = 0.041) were significantly overrepresented in AD-LOY patients (Fig 2C and D).

To analyze the association between LOY and the VAFs of MD-CH, LD-CH and UD-CH variants, we stratified the cohort according to the chromosome Y status (Fig 3A). UD-CHs were analyzed separately in AD and CTRL group (Fig 3B). If several variants of the same CH type were detected in a subject, only the one with the highest VAF was included in the analysis. The VAFs of MD-CH variants were significantly lower in LOY subjects (adjusted *P* = 0.020). The distributions of VAFs of LD-CH and UD-CH were similar in LOY and ROY individuals (Fig 3A). Although UD-CHs were enriched in AD-LOY group (Fig 2C), there were no significant differences in the distribution of their VAFs between AD-LOY and AD-ROY patients, similarly to the CTRLs (Fig 3B). The tendency of VAFs of MD-CH variants to be lower in the LOY group is also visible on Fig 4, where the percentages of cells (2xVAF) with different CH types are shown according to the %LOY in the analyzed cell fractions and the chromosome Y status on the subject level. Whereas MD-CH variants in LOY subjects tend to cluster on the lower part of the x-axis (Fig 4A), LD-CH and UD-CH variants are distributed across the entire scale, both in LOY and ROY group (Fig 4B and C).

**Table 2.  Recurrently affected genes and subjects with cell-type-specific CH variants.**

| CH type | Gene | LOY | ROY |
|---------|------|-----|-----|
| (A) | | | |
| MD-CH | *TET2* | 2 | 3 |
| MD-CH | *DNMT3A* | 1 | 2 |
| MD-CH | *SRSF2* | 0 | 2 |
| LD-CH | *KMT2D* | 2 | 0 |

| Cell type | LOY | ROY |
|-----------|-----|-----|
| (B) | | |
| CD4 | 2 | 4 |
| NK | 11 | 10 |
| MYEL | 7 | 4 |
| NK/MYEL | 19 | 6 |
| CD4/NK/MYEL | — | 2 |

**(A)** Number of subjects with CH variants affecting known myeloid (MD-CH) and lymphoid (LD-CH) driver genes recurred in LOY and ROY groups; **(B)** number of subjects with cell-type-specific CH variants: MD-CH, LD-CH, and unknown driver genes (UD-CH).

## Cell-type specific assessment of CH variants

To explore the cell-type specificity of CH variants, we assessed sharing of variants between sorted cell types (Table 2B). Two protein-truncating variants (MD-CH affecting *TET2* in ROY CTRL GK008 and LD-CH affecting *NFE2* in ROY CTRL GK017), predicted to have high oncogenic potential based on COSMIC and/or ClinGen-CGC-VICC guidelines (Horak et al, 2022), were detected in all three cell types. Both showed markedly lower VAFs in CD4⁺ T cells (0.05 and 0.01, respectively; Table S5) compared with NK and myeloid cells, where VAFs ranged from 0.10 to 0.17. Since this pattern suggested early clonal origin, likely at the level of hematopoietic stem or progenitor cells, with preferential expansion in NK and myeloid lineages, both variants were retained in the CH analysis as post-zygotic. Shared variants between NK and myeloid cells were observed in 25 subjects (19 LOY, 6 ROY), suggesting a common clonal origin with lineage-specific expansion. CH variants exclusive to NK cells were found in 21 subjects (11 LOY, 10 ROY). In contrast, CD4⁺ T cell-specific variants were rare, detected in only six individuals (4 ROY, 2 LOY).

The analysis of association between CH and LOY showed the total of 39 known MD/LD drivers, with 13 that possibly coexist with LOY: *ASXL1*, *DNMT3A*, *BRCA1*, *MED12*, *KMT2D* and *ATM* in NKs and *TET2*, *ASXL1*, *DNMT3A*, *KMT2D*, *BRCA1*, *ATM*, and *MGA* in myeloid cells (Fig 4A and B). Similarly, we found 65 and 67 UD-CH events in NK/myeloid cells, respectively, possibly co-occurring with LOY. The effect of LOY on total CH burden (a sum of VAFs of all post-zygotic variants detected in a sample) is presented in Fig 5. A significant Spearman correlation ($\rho = 0.52$, adjusted $P = 0.00041$) was observed in myeloid cells, but not in CD4⁺ T or NK cells, suggesting that LOY and CH tend to co-occur in granulocytes and monocytes, and LOY may be the primary driver of CH in myeloid lineage. On the contrary, CH in NK cells may be less frequently driven by LOY, as also supported by numerous variants with VAFs not proportional to %LOY (Fig 4, Table S5).

At the individual level, CH displays considerable complexity (Table S5). In the LOY group, clonal expansion appears to be driven by LOY in NK and myeloid cells in selected cases (e.g., M454), often accompanied by LD-CH or UD-CH. VAFs correspond to LOY levels in some individuals but not in others (e.g., GK037, M428), suggesting that LOY is not always the primary driver of CH. Other cases, like KAD054, further highlight the diversity in clonal architecture and variable contributions of LOY and post-zygotic variants across cell types. In ROY subjects, classical cases are observed where expansion is driven by oncogenic myeloid variants coexisting with variants of uncertain significance in lymphoid or unknown driver genes (e.g., subject M401). Some individuals (e.g., patient M394), exhibit NK-specific LD-CH variants with high oncogenic potential as a likely driver, alongside additional variants of other CH types. Donor GK008 showed a complex pattern with one specific MD-CH variant in myeloid cells and additional UD-CH variant(s) specific to CD4⁺ T or NK cells.

## Functional analysis of UD-CHs

Unknown driver variants were found in 48 subjects and may contribute to clonal hematopoiesis, especially in individuals without deleterious variants in known canonical drivers or LOY. As no recurrently mutated unknown driver genes were identified, we focused on their molecular functions and associated biological processes. To reduce the possibility of inclusion of variants of uncertain significance (VUS), we analyzed only protein-truncating and highly deleterious missense variants (M-CAP > 0.025). Five main processes were identified: regulation of transcription by RNA polymerase II and DNA-templated transcription (12 genes: *HBP1*, *ABRA*, *CTBP2*, *ETV3*, *ZNF100*, *HNF1A*, *BMAL1*, *MTA2*, *NKAP*, *RBM10*, *MED12L*, *ADORA3*), regulation of splicing or transcriptional and post-transcriptional regulation of RNA (three genes: *DCDC2*, *PLEKHH1*, *CFAP96*), Rho GTPases and small GTPase-mediated signal transduction connected to cytoskeleton remodeling (six genes: *SYDE1*, *DMPK*, *MYH3*, *ABRA*, *RRAD* and *HEG1*), calcium homeostasis (six genes: *HEG1*, *RRAD*, *RYR1*, *GRIN2B*, *HTR5A*, *ATP2B3*) and ubiquitin-protein transferase pathways (two genes: *RBBP6* and *PDZRN3*).

To identify biological processes potentially disrupted by detected post-zygotic variants in all driver genes, we performed Gene Set Enrichment Analysis (GSEA). The analyzed cohort was stratified according to the chromosome Y status and disease status, which resulted in four groups: CTRL-ROY, CTRL-LOY, AD-ROY and AD-LOY. Results from two gene ontology databases are shown in Table S6. In the AD-LOY group, significant enrichment ($Q < 0.05$) was observed for gene sets related to "immune system process" (NES = 2.62, q = 10-4) and "hemopoiesis" (NES = 2.49, $Q < 10^{-3}$), driven by genes including *ASXL1* and *ATM*, suggesting perturbation of hematopoietic regulation and immune response. The AD-ROY group showed enrichment in regulatory gene sets, specifically targets of the miR-29 family (NES = 2.21, q < 0.01) and *MYC* transcription factor (NES = 2.07, q = 0.039). This enrichment was associated with mutations in epigenetic modifiers *DNMT3A* and *TET2*, indicating

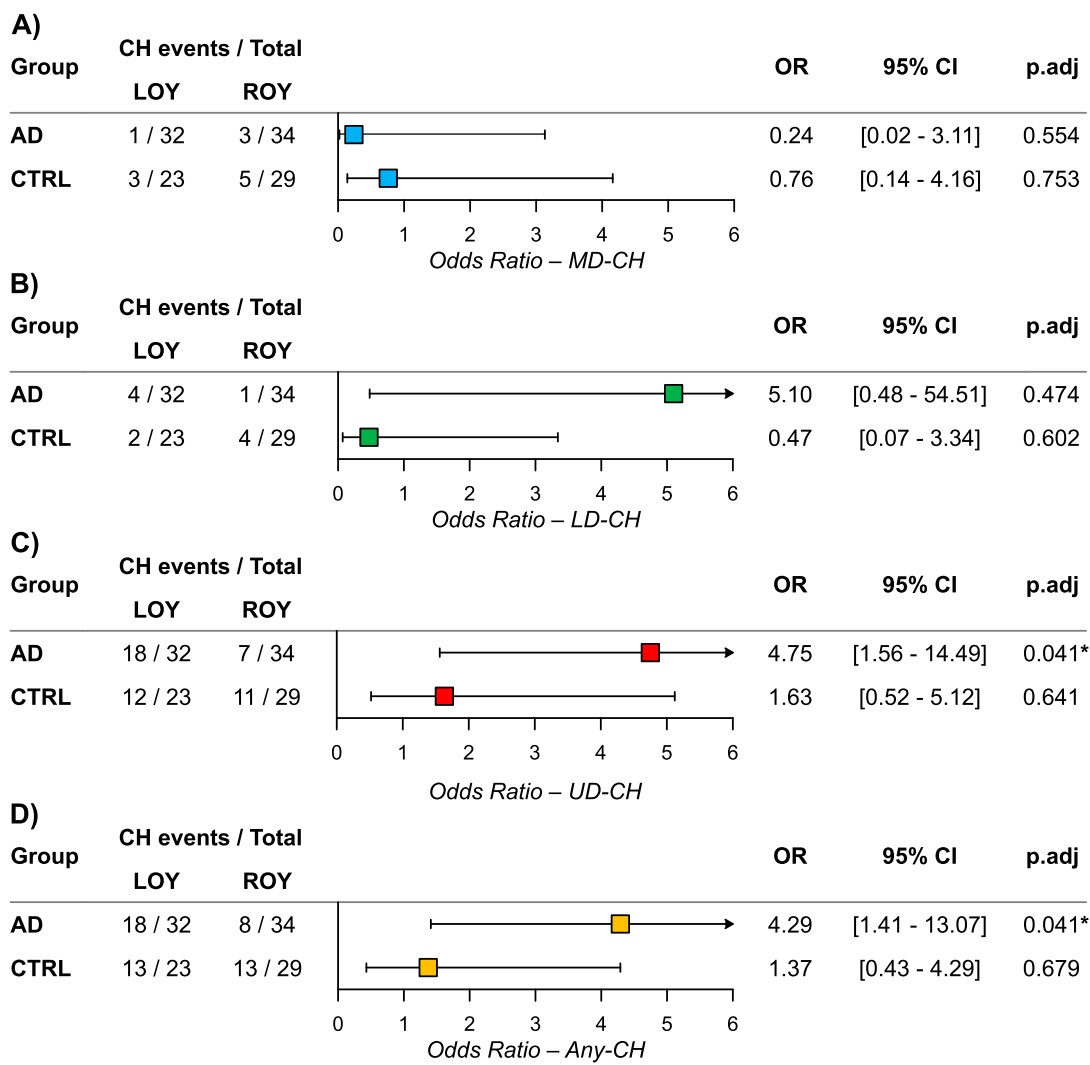

**Figure 2. Association between loss of Y chromosome (LOY) and CH types.**
**(A, B, C, D)** LOY and MD-CH; (B) LOY and LD-CH; (C) LOY and UD-CH; (D) LOY and all CH combined (Any-CH). Proportions of subjects with each CH type were compared between LOY and ROY groups within AD patients (AD) and controls (CTRL) using logistic regression adjusted for age and age$^2$. Odds ratios (ORs) with 95% confidence intervals (CI) are shown. P-values were Benjamini-Hochberg adjusted for eight comparisons (P.adj), with significant adjusted P < 0.05 marked by an asterisk. Arrows indicate CIs extending beyond axis limits.

dysregulation of epigenetic and transcriptional networks. In the CTRL-ROY group, enrichment was significant for "leukemia" and "acute leukemia" terms (NES = 2.12, Q = 0.03), driven by key myeloid malignancy-associated genes *DNMT3A*, *IDH1*, *SRSF2*, and *TET2*. This points towards a disruption of hematopoietic stem cell function predisposing to myeloid neoplasia. We did not detect any significantly enriched gene sets in the CTRL-LOY group. The analysis of mRNA expression of the 192 UD-CH genes in GeneCards database showed that the vast majority of them were clearly and highly expressed in immune cells. *SYTL5*, *TRIM71*, *IL1RAPL1*, *MAGEC1*, *SALL3*, *OR5K1*, *PCDHA12* expressed at intermediate-low level, whereas *FAM237A*, *OR5T1*, *AMER3* were at low levels, and *PRAMEF17* had no expression detectable above background in hematopoietic cells.

## Discussion

CH and LOY are common events in elderly; however, their co-occurrence is a matter of debate. We used FACS to isolate multiple hematopoietic cell lineages from each participant combined with deep WES. This approach enables robust detection of post-zygotic variants restricted to specific cell types and provides statistically meaningful LOY-point mutation co-occurrence estimates even in a modest-sized cohort, demonstrating the feasibility of scaling up this strategy. Querying public repositories with Ensembl Variant Effect Predictor (McLaren et al, 2016), we found that 57% of post-zygotic variants in our combined WES dataset are novel, 11% match catalogued post-zygotic only entries, and 32% bear mixed germline/post-zygotic annotations, underscoring our multi-lineage design as

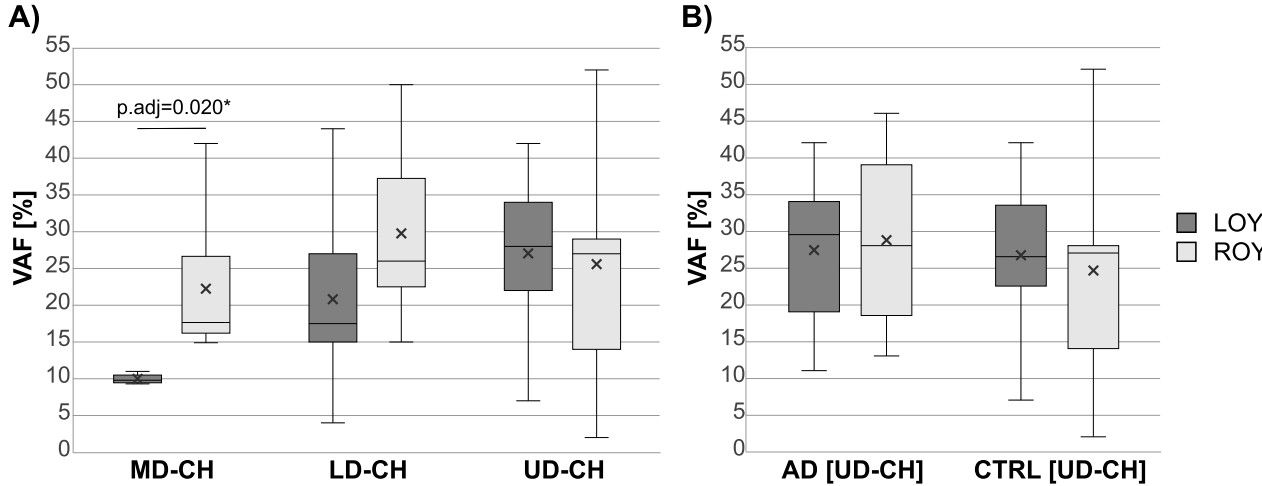

**Figure 3. VAFs of different types of CH variants versus loss of Y chromosome (LOY).**
The boxplots present the distribution of VAFs of variants detected in LOY and ROY subjects. **(A, B)** VAFs of MD-CH, LD-CH and UD-CH variants in the cohort stratified according to the subject's chromosome Y status only (AD patients and controls combined); (B) VAFs of UD-CH variants in AD patients and controls. In case if more than one variant of the same CH type was detected in a subject, only the variant with the highest VAF was included in the analysis. The differences between distributions of VAFs in LOY and ROY subjects were tested using the Mann-Whitney *U* test. The *P*-values were adjusted for multiple testing using Benjamini-Hochberg method assuming five tests. Statistically significant adjusted *P*-value was marked with an asterisk. The line inside the boxplot represents the median; the cross inside the box represents the mean; the upper border of the box represents Q3; the lower border of the box represents Q1; whiskers extend from minimum to maximum. VAF, variant allele frequency; AD, Alzheimer disease; LOY, loss of Y chromosome; ROY, retention of Y chromosome; CH, clonal hematopoiesis; MD, myeloid driver; LD, lymphoid driver; UD, unknown driver; Q3, third quartile; Q1, first quartile.

strongest point of the study. This support extending this framework to additional cell types and considerably larger cohort to refine variant catalogues and elucidate lineage-specific architecture.

We show that LOY infrequently co-occurs with variants in established drivers of myeloid and lymphoid malignancies, both in AD and CTRL cohorts (Fig 2A and B), suggesting that LOY alone does not drive the accumulation of highly pathogenic changes. We found altogether 39 sequence variants for known MD/LD drivers and maximally 14 (35%) of them could co-exist with LOY in the same hematopoietic clone (Fig 4A and B). Additionally, the VAFs of detected MD-CH variants were significantly lower in LOY subjects (Figs 3A and 4A), further supporting the lack of co-occurrence (Kamphuis et al, 2023; Mas-Peiro et al, 2023) or even mutual exclusivity of LOY and CHIP (Brown et al, 2023). The overall frequency of MD-CH in our cohort was 10.2%, irrespective of AD or LOY status, and recurrently affected MD genes were *TET2*, *DNMT3A*, and *SRSF2*, which is consistent with the literature (Genovese et al, 2014; Jaiswal et al, 2014, 2017; Niroula et al, 2021; Dawoud et al, 2023; Woo et al, 2024). However, when stratifying the cohort according to disease status, we found that only 6.1% of AD patients had variants in myeloid drivers in comparison to 15.4% of CTRLs. Although this difference did not reach statistical significance, most probably due to the small cohort size, it is consistent with other studies, which report that CHIP is associated with reduced risk of AD (Bouzid et al, 2023). It should also be stressed that even though AD patients in our cohort are ~10 yr older than controls, we do not see the accumulation of MD-CH with age in this group, which is in contrast to reports showing higher frequency of CHIP in older individuals (Niroula et al, 2021; Kamphuis et al, 2023), and supporting the hypothesis of CHIP being the factor reducing the AD risk. However,

the small sample size in our study remains a key limitation that could influence both the strength and direction of the observed associations. Indeed, recent findings indicate that CHIP mutations are more common in AD patients compared with age-matched controls and confer a higher risk of AD (Naito et al, 2025 *Preprint*; Choi et al, 2025 *Preprint*). Therefore, the results of our study should be validated by further research on larger cohorts.

Similar to MD-CH, there was also no enrichment of LD-CH in LOY individuals; however, the OR for AD cohort was higher than for UD-CH, though not statistically significant, likely due to the rarity of these events and small cohort size (Fig 2B). The distributions of VAFs of LD-CH variants were similar in LOY and ROY groups (Fig 3A). Variants in genes driving lymphoid malignancies are usually classified as variants of uncertain significance (VUSs) with lower pathogenicity scores, and similar distribution of VAFs in LOY and ROY subjects could further support the hypothesis that LOY is not associated with highly pathogenic changes, but rather is followed by accumulation of VUSs. In contrast to MD-CH and LD-CH, UD-CH and Any-CH variants were significantly overrepresented in AD patients with LOY (Fig 2C and D), although there were no differences in the distributions of their VAFs between LOY and ROY individuals, both in AD patients and CTRLs (Fig 3A and B). Additionally, comparison of the variants identified in our study revealed overlap with previously reported LOY-associated transcriptional effects (LATEs) genes: 18 UD-CH and 2 LD-CH genes had been reported to be dysregulated in myeloid cells of LOY subjects with AD (Dumanski et al, 2021; Jąkalski et al, 2025), whereas 51 UD-CH, 4 LD-CH, and 2 MD-CH genes corresponded to hypomethylated genes found in granulocytes and monocytes of AD-LOY patients (Jąkalski et al, 2025). These might suggest a role for LOY as a primary driver of these non-pathogenic passenger mutations in

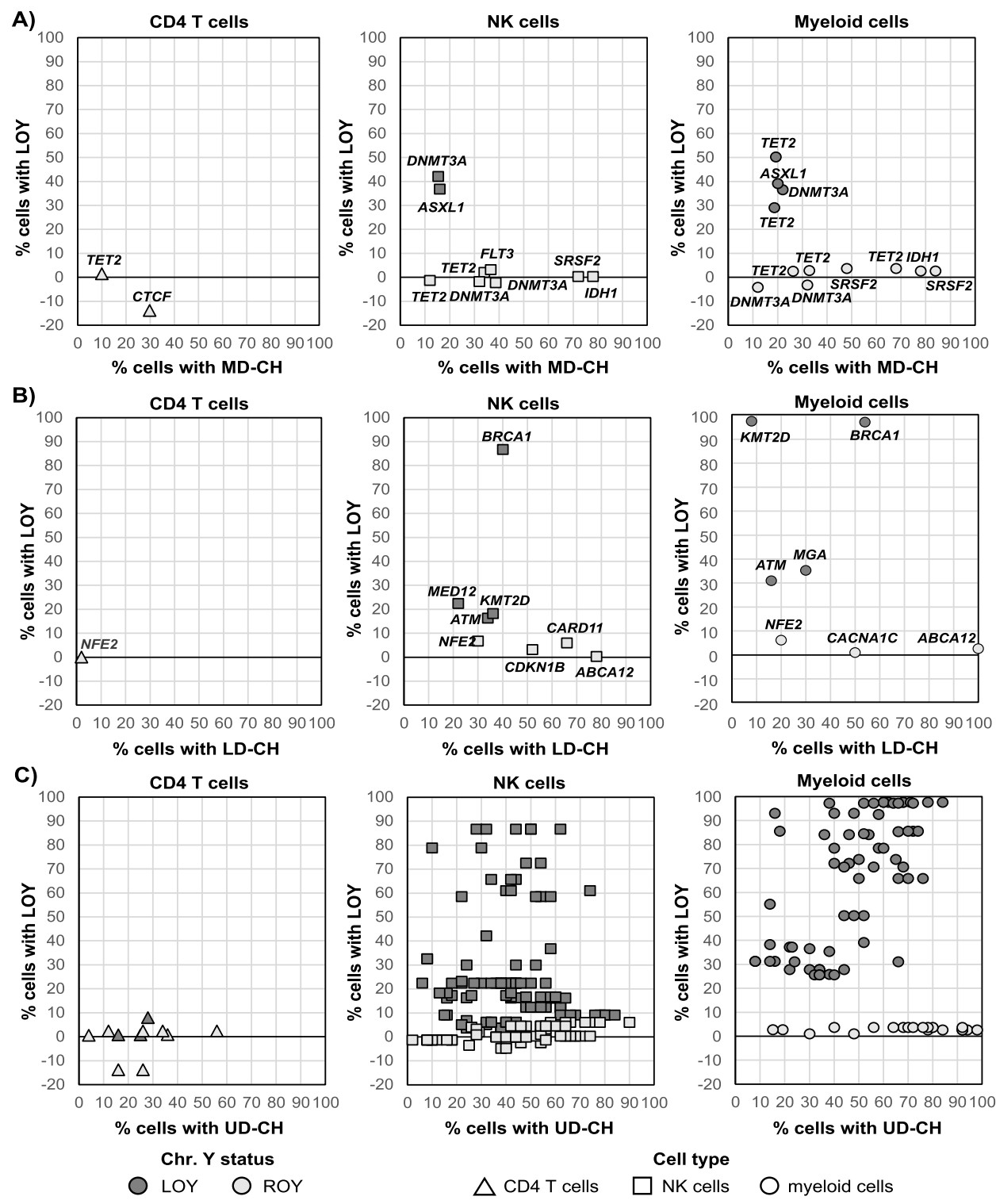

**Figure 4. CH versus loss of Y chromosome (LOY) across analyzed cell types.**
Scatterplots display the percentage of cells carrying each CH variant type (x-axis) versus the corresponding percentage of cells with LOY (y-axis). **(A, B, C)** MD-CH versus LOY in CD4⁺ T cells, NK cells and myeloid cells; (B) LD-CH versus LOY in the same cell types; (C) UD-CH versus LOY in the same cell types. Each point represents one variant detected in one subject. All variants detected in a given subject were shown. Gray background encodes Y chromosome status on the subject level (LOY ≥ 15% in any cell fraction); point shape denotes cell type. Gene symbols are annotated for MD-CH and LD-CH variants.

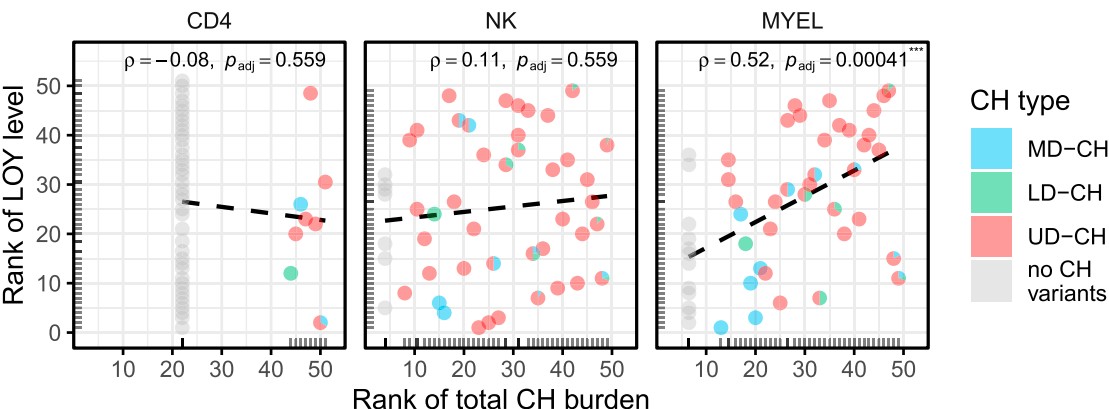

**Figure 5. Total CH burden versus loss of Y chromosome across cell types.**
For CD4⁺ T-cells, NK-cells, and myeloid cells in respective facets, these scatterplots illustrate the strength and direction of the monotonic association between total CH burden (defined as sum of VAFs of all post-zygotic variants detected in a sample; x-axis) and loss of Y chromosome level (y-axis). Axes are equal-scaled, variables are converted to ranks within their respective cell-type, and bottom and left rugs indicate marginal rank distributions. Each subject is shown as a pie glyph depicting the proportional contribution of MD-CH (blue), LD-CH (green) and UD-CH (red) variants. Samples lacking detectable CH are rendered as solid grey pies. Dashed black lines are least-squares fits to the ranked data, visually representing Spearman's correlation coefficient ρ and its BH adjusted *P*-values.

the AD cohort. It must be underscored that LOY at cellular fractions ≥30% has been associated with MD-CH, LD-CH, and UD-CH in the general population (Dawoud et al, 2023). However, determining whether LOY and CHIP truly co-occur within the same cell requires single-cell level analysis. In our cohort, LOY was detected in 15–97% of cells across 83 samples from 54 individuals (median for NK—16.4%, and for myeloid cells—38.2%). Given that the median VAF of mutations detected in NKs was 22%, and in myeloid cells 23% in LOY subjects, it is not possible to reliably infer the co-occurrence of LOY and CHIP within the same cell.

We also analyzed the UD-CHs for their pathogenicity and potential to drive CH. Overall, we uncovered 192 UD genes in 48 subjects, and these variants were predominantly observed in myeloid and NK cells. Twenty percent of UDs represent protein truncating mutations and these likely perturb immune homeostasis and hematopoietic regulation in which these proteins participate. The analysis of gene expression also showed that the vast majority of UD-CH genes are expressed in hematopoietic cells, supporting their potential relevance for CH. Further analysis of biological processes and molecular functions showed that in the AD-LOY cohort, post-zygotic mutations are significantly enriched in gene sets related to hematopoiesis, suggesting that LOY in combination with CH may perturb immune homeostasis. Moreover, shared UD-CHs were frequent between NK and myeloid cells especially in LOY subjects. Thus, LOY can act as a clonal driver in myeloid cells (and partially also in NKs), but its role is not universal and clonal expansions frequently occur independently of LOY. It is important to consider that some of the observed UD-CHs may represent passengers, expanding not through their own selective advantage but by clonal expansions initiated by LOY or other drivers. In this context, LOY may play a more central initiating role, potentially leading to the accumulation of additional mutations.

Consistent with the above, we demonstrate a relationship between the total burden of observed CH variants and LOY in the myeloid lineage, primarily driven by UD-CHs (Fig 5). This supports the notion that CH is not driven by a restricted set of genes but may

involve a considerably broader range of mutations affecting regulatory processes in hematopoietic cells and this issue should be studied using sorted subpopulations of blood cells rather than using bulk DNA derived from all leukocytes. This could help in refining CH beyond already known driver genes. The lack of gene-level recurrence, particularly among UD-CHs, highlights the heterogeneous nature of CH and suggests that rare, possibly individual-specific, mutations may contribute to clonal expansion. However, the functional relevance of many detected variants remains uncertain. Distinguishing true drivers from passengers remains a major challenge and requires larger cohorts and extensive functional validation. Overall, our approach to analysis of clonal expansion using deep WES in sorted cells suggests a complex picture of CH with potentially many more additional drivers, pointing to a still largely unexplored heterogeneity of CH-related sequence variants that may or may not be coexisting with LOY. However, given the relatively small size of our cohort and lack of functional studies, conclusions about the biological significance of UD-CH variants should be viewed as preliminary. The enrichment of UD-CHs in LOY-positive individuals may reflect a biological signal but could also result from increased genomic instability, rather than causality. Many of the variants identified reside in genes without established role in hematopoiesis or Alzheimer's disease and were prioritized based on in silico predictions alone. Thus, whereas our deep WES approach in sorted cells reveals a complex and underexplored CH mutational landscape, further studies are essential to validate the functional roles of UD-CH variants and their possible contributions to disease.

## Materials and Methods

### Study participants

Sixty-six AD patients were recruited at the Adult Psychiatry Clinic of the University Clinical Center in Gdańsk, the Clinic of Internal

Diseases and Gerontology of the Jagiellonian University in Kraków and the Memory Clinic at Uppsala University Hospital. Fifty-two controls (CTRLs), recruited from the general population of Gdańsk, Kraków, and the EpiHealth study in Uppsala, met inclusion criteria of being over 65 yr old with no self-reported history of cancer or dementia. Subjects were recruited between 2015 and 2022. Written informed consent was obtained from all participants. The study was performed in accordance with the Declaration of Helsinki and approved by the Independent Bioethics Committee for Research at the Medical University of Gdańsk (decision number NKBBN/564/2018 with amendments), the Bioethical Committee of the Regional Medical Chamber in Kraków (decision number 6/KBL/OIL/2014 with amendments), and the Uppsala Regional Ethical Review Board (Regionala Etikprövningsnämnden in Uppsala [EPN]: Dnr 2005-244, Ö48-2005; Dnr 2015/092; Dnr 2015/458; Dnr 2015/458/2, the latter with update from 2018).

## Isolation of target leukocyte subpopulations and determination of LOY

Target leukocyte subpopulations (CD4+ T cells, NK cells, granulocytes, and monocytes) were isolated from peripheral blood with FACS, following two published protocols, presented in Dumanski et al (2021) and Wójcik et al (2024). The proportion of cells with LOY (%LOY) in each cell type was determined at the DNA level using mLRR-Y values from Illumina SNP arrays (Forsberg et al, 2014) and/or droplet digital PCR (ddPCR) to study the *AMELX*/*AMELY* genes (Danielsson et al, 2020). Cell subpopulations with ≥15% of cells lacking the Y chromosome were classified as LOY, and subjects were assigned to the LOY group if any target subpopulation met this criterion.

## WES and post-zygotic variants identification

WES was outsourced to Macrogen Europe. Libraries were prepared using the SureSelect Human All Exon V7 kit (Agilent Technologies) with a low-input protocol and sequenced on the NovaSeq6000 Illumina Platform using 2 × 150 bp paired-end reads to achieve an average on-target coverage of 150X. The sequencing coverage and quality statistics for each sample are summarized in Table S1.

Raw FASTQ files were assessed using the FastQC tool (http://www.bioinformatics.babraham.ac.uk/projects/fastqc) and processed with Trim Galore! (v0.6.7) (http://www.bioinformatics.babraham.ac.uk/projects/trim_galore) to remove Illumina-specific adapter sequences and poor-quality reads, when necessary. The reads were mapped to the human reference genome (hg38) using Burrows-Wheeler Alignment tool (BWA-MEM) (Li & Durbin, 2009). Reads were converted to uBAM format and read groups were extracted from the raw data. Post-zygotic single nucleotide variants (SNVs) and small indels were identified according to GATK4 best practices (Van der Auwera and O'Connor, 2020), and variant calling was performed separately using Platypus v0.8.1.1. A list of 97 myeloid and 335 lymphoid driver genes was curated based on current CH research (Tables S2 and S3) (Beauchamp et al, 2021; Niroula et al, 2021; Ljungström et al, 2022; Pich et al, 2022; Brown et al, 2023; Dawoud

et al, 2023; Vlasschaert et al, 2023) with all other protein-coding genes categorized as unknown drivers.

SNVs and small indels were identified using Platypus Variant Caller v0.8.1.1 (Rimmer et al, 2014) in tumor-only mode. Variants in reads with poor mapping quality (<30), and variants supported by high-quality bases (≥30) in fewer than five reads were excluded from the analysis. Detected variants were functionally annotated using ANNOVAR (Wang et al, 2010). In-house R script was used to compare variants identified across cell types from a given subject to designate them as germline (if present in all cell types) or post-zygotic (if present in one or more, but not all cell types). To facilitate detection of post-zygotic changes we included variants flagged as PASS and alleleBias, and filtered by their frequency in the general population, retaining only those with a minor allele frequency (MAF) ≤ 0.01 across all gnomAD populations ("popmax") or not listed in gnomAD (v2.1.1) (Chen et al, 2024).

Only variants located in exons (frameshift insertions/deletions, nonsense, and missense variants) were included in the analysis. To minimize the possibility of inclusion of germline variants with imbalanced VAFs or sequencing artefacts, we focused on changes classified as post-zygotic and occurring only once in the cohort (singleton variants), except for highly oncogenic variants in myeloid driver genes, which were included even if occurred in more than one subject.

For MD-CH variants, we retained only those with a sequencing depth ≥20, at least three alternate reads with balanced forward/reverse strand coverage, and tissue allele frequency ≥0.01.

For LD-CH and UD-CH variants, more stringent quality criteria were applied, retaining only those with sequencing depth ≥20, at least five alternate reads with balanced forward/reverse strand coverage, and a tissue allele frequency ≥0.02.

Filtered variants were manually inspected across all analyzed cell types using the Integrative Genomics Viewer (IGV) (Robinson et al, 2017) to confirm their post-zygotic origin and determine cell-type specificity. If a variant was detected in one cell type but not in others, BAM files from all cell types were examined in IGV to assess whether Platypus may have missed it. In such cases, the variant was considered present in a given cell type if it was supported by at least five alternate reads with high base quality (≥60), and the site had a total read depth of at least 20 with mapping quality ≥60.

## Statistical analysis

Fisher's exact test and logistic regression adjusted for age and age[2] was used to assess the association between LOY and CH types in AD and CTRL donors, with the association strength expressed as OR with 95% CI. Differences in the distribution of age, %LOY and VAFs were evaluated using the Mann-Whitney $U$ test. Spearman's rank-based correlation was calculated within each cell type between the total CH burden (sum of variants' VAFs) and the LOY level, and visualized with a least-squares line fitted to the ranked values of these two variables. All $P$-values were adjusted for multiple comparisons using Benjamini-Hochberg (BH) method. Fisher's

exact test, logistic regression and the Mann-Whitney $U$ tests were performed with TIBCO Statistica v13.3 (TIBCO Software Inc.), whereas Spearman's rank-based correlation analysis and GSEA were carried out using R v4.4 (R Core Team, 2025).

## GSEA and mRNA expression analysis

GSEA was conducted using clusterProfiler v4.14.6 (Yu et al, 2012) with default settings, except that the minimum gene set size was set to five. Genes with post-zygotic SNVs were ranked by classification, frequency and predicted functional impact. The analysis integrated multiple annotation databases, including Gene Ontology (Ashburner et al, 2000) Biological Process (GO_BP) and Molecular Signatures Database (Subramanian et al, 2005) (MSigDb 2024.1.hs). Gene sets with permutation-based BH adjusted $P$-values < 0.05 were considered significantly enriched. We also analyzed the mRNA expression of the 192 UD-CH genes, focusing on hematopoietic cells, using data from GeneCards database (www.genecards.org) (Stelzer et al, 2016). A gene was classified as expressed above background if in GeneCards any mRNA level in any immune cell type (bone marrow, whole blood, white blood cells, lymph node, or thymus) shows $(100 \times FPKM)$ $1/2 > 1$ for bulk RNA-seq or (intensity) $2/3 > 1$ for microarray data, or if protein expression in any blood and immune tissue (serum, plasma, monocyte, neutrophil, B-lymphocyte, T-lymphocyte, CD4$^+$ T cells, CD8$^+$ T cells, NK cells, PBMCs, platelets, lymph node, tonsil, bone marrow stromal cell, or bone marrow mesenchymal stem cell) was $\log_{10}$ (ppm) > –1.

# Data Availability

All raw sequencing data generated in this study have been submitted to the EGA—European Genome-Phenome Archive (EGA: https://ega-archive.org/) under accession number EGAS00001008234.

# Supplementary Information

# Acknowledgements

We would like to thank all the patients and healthy controls for sample contribution and information provided in the questionnaires. We are grateful to Dr. Magdalena Koczkowska and Dr. Jakub Mieczkowski for the scientific support and advice. This study was supported by grant from Foundation for Polish Science (MAB/2018/6) to JP Dumanski and A Piotrowski, as well as grants from Vetenskapsrådet, Cancerfonden, Alzheimerfonden and Hjärnfonden to JP Dumanski.

## Author Contributions

E Rychlicka-Buniowska: conceptualization, resources, data curation, formal analysis, investigation, visualization, methodology, and writing—original draft, review, and editing.
D Sarkisyan: formal analysis, visualization, methodology, and writing—original draft, review, and editing.
M Horbacz: data curation, formal analysis, visualization, methodology, and writing—review and editing.
B Bruhn-Olszewska: resources, investigation, and writing—review and editing.
K Dręzek-Chyła: resources, investigation, and writing—review and editing.
M Koszyński: resources, investigation, and writing—review and editing.
H Davies: resources, investigation, and writing—review and editing.
U Juhas: resources, investigation, and writing—review and editing.
M Wójcik-Zalewska: resources, investigation, and writing—review and editing.
A Klich-Rączka: resources, methodology, and writing—review and editing.
L Kilander: resources, methodology, and writing—review and editing.
M Ingelsson: resources, methodology, and writing—review and editing.
K Bukowska-Strakova: resources, investigation, and writing—review and editing.
K Węglarczyk: resources, investigation, and writing—review and editing.
M Siedlar: resources, methodology, and writing—review and editing.
J Baran: resources, methodology, and writing—review and editing.
J Ryś: resources, methodology, and writing—review and editing.
A Piotrowski: resources, formal analysis, funding acquisition, methodology, and writing—review and editing.
N Filipowicz: conceptualization, resources, formal analysis, supervision, investigation, methodology, and writing—original draft, review, and editing.
JP Dumanski: conceptualization, resources, formal analysis, supervision, funding acquisition, methodology, and writing—original draft, review, and editing.

## Conflict of Interest Statement

The authors declare that they have no conflict of interest.

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
