## [Reviewer comments · Life Science Alliance]

Loss of Y chromosome in Alzheimer's patients co-occurs with somatic mutations beyond CHIP drivers

Edyta Rychlicka-Buniowska, Daniil Sarkisyan, Monika Horbacz, Bożena Bruhn-Olszewska, Kinga Drężek-Chyła, Mikołaj Koszyński, Hanna Davies, Ulana Juhas, Magdalena Wójcik-Zalewska, Alicja Klich-Rączka, Lena Kilander, Martin Ingelsson, Karolina Bukowska-Strakova, Kazimierz Weglarczyk, Maciej Siedlar, Jarosław Baran, Janusz Ryś, Arkadiusz Piotrowski, Natalia Filipowicz, and Jan Dumanski

DOI: <https://doi.org/10.26508/lsa.202503533>

Corresponding author(s): Jan Dumanski, Uppsala University and Edyta Rychlicka-Buniowska, Gdańsk Medical University

Review Timeline:

Submission Date:	2025-10-13
Editorial Decision:	2025-10-14
Revision Received:	2025-10-22
Editorial Decision:	2025-11-11
Revision Received:	2025-11-14
Accepted:	2025-11-18

Scientific Editor: Tim Fessenden

Transaction Report:

Please note that the manuscript was previously reviewed at another journal and the reports were taken into account in the decision-making process at *Life Science Alliance*. Since the original reviews are not subject to Life Science Alliance's transparent review process policy, the reports and author response cannot be published.

October 14, 2025

Re: Life Science Alliance manuscript #LSA-2025-03533-T

Jan P Dumanski

Department of Immunology, Genetics and Pathology and Science for Life Laboratory, Uppsala University, Uppsala, Sweden

Dear Dr. Dumanski,

Thank you for transferring your manuscript entitled "Loss of Y chromosome in Alzheimer disease patients co-occurs with clonal hematopoiesis defined by post-zygotic point mutations outside canonical CHIP driver genes" to Life Science Alliance. In accordance with our offer to consider this work and our prior correspondence, we invite you to submit a revised manuscript.

Thank you for this interesting contribution to Life Science Alliance. We are looking forward to receiving your revised manuscript.

Sincerely,

B. MANUSCRIPT ORGANIZATION AND FORMATTING:

November 11, 2025

RE: Life Science Alliance Manuscript #LSA-2025-03533-TR

Prof. Jan P Dumanski
Uppsala University
Department of Immunology, Genetics and Pathology
Box 815
Uppsala 751 08
Sweden

Dear Dr. Dumanski,

Thank you for submitting your revised manuscript entitled "Loss of Y chromosome in Alzheimer's patients co-occurs with somatic mutations beyond CHIP drivers". As you will see, the original reviewers are now satisfied with no further requests. We would be happy to publish your paper in Life Science Alliance pending final revisions necessary to meet our formatting guidelines.

- Please add the X and Bluesky handles of your host institute/organization, as well as your own and/or one of the authors, in our system.
- It is recommended to exclude figures from the manuscript text.
- Please upload your main and supplementary figures as single files.
- Please add your main, supplementary figure, and table legends to the main manuscript text after the references section.
- Tables can be included at the bottom of the main manuscript file or be sent as separate files in editable .doc or Excel format.
- Please rename the Data Access section to Data Availability.
- Please label the Author Contribution section.
- Please add a callout for Figure 4C to your main manuscript text.
- The abstract nicely summarizes the background and main findings, but makes extensive use of abbreviations. We suggest reducing the use of these abbreviations to render the abstract more reader-friendly.

A. FINAL FILES:

B. MANUSCRIPT ORGANIZATION AND FORMATTING:

Thank you for your attention to these final processing requirements. Please revise and format the manuscript and upload materials as soon as you are able.

Sincerely,

November 18, 2025

RE: Life Science Alliance Manuscript #LSA-2025-03533-TRR

Prof. Jan P Dumanski
Uppsala University
Department of Immunology, Genetics and Pathology
Box 815
Uppsala 751 08
Sweden

Dear Dr. Dumanski,

Thank you for submitting your Research Article entitled "Loss of Y chromosome in Alzheimer's patients co-occurs with somatic mutations beyond CHIP drivers". We note your attention to the minor issues we raised on our previous decision letter. It is a pleasure to let you know that your manuscript is now accepted for publication in Life Science Alliance. Congratulations on this interesting work.

DISTRIBUTION OF MATERIALS:

Again, congratulations on a very nice paper. I hope you found the review process to be constructive and are pleased with how the manuscript was handled editorially. We look forward to future exciting submissions from your lab.

Sincerely,
